# Treatment of Bovine Leptospirosis with Enrofloxacin HCl 2H_2_O (Enro-C): A Clinical Trial

**DOI:** 10.3390/ani12182358

**Published:** 2022-09-09

**Authors:** Jesús Mendoza Bautista, Melissa Aranda Estrada, Lilia Gutiérrez Olvera, Reyes Lopez Ordaz, Héctor Sumano López

**Affiliations:** 1Department of Physiology and Pharmacology, National Autonomous University of Mexico (UNAM), Av. Universidad 3000, Mexico City 04510, Mexico; 2Department of Agricultural and Animal Production, Autonomous Metropolitan University (UAM), Calz. del Hueso 1100, Mexico City 04960, Mexico

**Keywords:** cows, clinical trial, leptospirosis, enrofloxacin HCl·2H_2_O, enro-C

## Abstract

**Simple Summary:**

Unlike standard enrofloxacin, its crystal solvate enrofloxacin HCl-2H_2_O (enro-C) exhibits favorable pharmacokinetics and PK/PD ratios compatible with its use as a treatment during the clinical phase of leptospirosis in cattle. Hence, a clinical trial was conducted in cattle located either in the highlands (HL) or the tropics (TL). A high dose of 15 mg/kg day IM of enro-C administered over 5 days was compared against the conventional treatment with streptomycin/penicillin-G. The cows treated with enro-C became PCR negative: 87.5% and 78.94% on day 5, and 92.85% and 94.73% on day 28 (in the HL and TL, respectively). For streptomycin/penicillin-G, the same values were 65.45% and 70.90% on day 5 and 73.68% in both scenarios on day 28. In both groups, the microagglutination titers dropped on day 28, and gestation rates were statistically indistinguishable between the groups. This is the first report of successful treatment with fluoroquinolone in treating bovine leptospirosis.

**Abstract:**

Pharmacokinetics/pharmacodynamics ratios of enrofloxacin HCl-2H_2_O (enro-C) in cows to treat bovine leptospirosis prompted this clinical trial in the highlands (HL) and the tropics (TL) in Mexico. In the HL, 111 Holstein-Friesian cows were included and 38 F1 Zebu–Holstein/Friesians in the TL. Affected cows were randomly divided into two treatment groups, both in the HL and TL. PCR and MAT tests were performed before and after treatment. Treatments in both groups were administered for 5 d with either IM injections of enro-C or streptomycin/penicillin-G. Reproductive performance data were gathered for 90 d. The cows treated with enro-C became PCR negative: 87.5% and 78.94% on day 5, 92.85% and 94.73% on day 28 (in the HL and TL, respectively). For streptomycin/penicillin-G, the same values were 65.45% and 70.90% on day 5, and 73.68% twice on day 28 in the HL and TL, respectively. In both groups and geographical settings, the MAT titers dropped on day 28 but remained above reference values usually considered negative. The gestation rates were: 86.53% and 79.06% and 88.88% and 87.5% for the HL and TL, either with enro-C or streptomycin/penicillin-G, respectively. This is the first report of successful treatment with a fluoroquinolone derivative in treating bovine leptospirosis with a high bacteriological cure rate.

## 1. Introduction

Leptospirosis is a zoonotic disease with a global distribution that affects domestic and wild animals. It has particular incidence in tropical and subtropical regions [1]. Approximately 20 species have been described as important, with more than 250 serovars [2]. The distribution and population patterns of infection can change as serovars adapt to different hosts in a given geographic area, as well as climatic and ecological changes [3,4]. In tropical countries, bovine leptospirosis is endemic, a situation favored by the high frequency of pluvial precipitation, warm temperatures, and soil pH, which are factors that guarantee the survival of the *Leptospira* genus [5,6]. The health and economic impact of leptospirosis is often underrated due to the complexity of establishing a rapid and accurate diagnosis, as leptospirosis often manifests with nonspecific symptoms. Additionally, there is considerable difficulty in isolating and culturing the pathogen, and the internationally accepted microscopic agglutination test (MAT) is challenging to interpret, particularly in the chronic phase of the disease [7].

Bovine leptospirosis is mainly caused by the *Leptospira interrogans* serovar Hardjo (Hardjobovis and hardjoprajitno). However, it was found that other serovars such as Pomona, Icterohaemorrhagiae, and Grippotyphosa are associated with clinical outbreaks of this disease [8]. The chronic presentation of leptospirosis in dairy cattle is associated with important economic losses due to abortions in the last third of gestation; the mummifications of fetuses or calving of weak or dead offspring; infertility; or reduction in the reproductive performance of the herd, increase in mastitis incidence, and decreased milk production [7,9]. Control of leptospirosis is based on environmental management, vaccination, and antibacterial therapy, while attempting to achieve a bacteriological cure, decrease the risk of transmission within the herd, and diminish losses due to reproductive problems. The efficacy of some antibacterial agents is variable, and they can often result in poor clinical and/or bacteriological cure rates [10,11]. A typical scenario is that some cows remain as chronic renal carriers, intermittently shedding leptospires, thus re-disseminating the disease [6,12]. The recommended treatments are based on the combination of penicillin and streptomycin or streptomycin–tetracycline combinations [13]. Additionally, ceftiofur has shown acceptable efficacy [14]. In some acute cases, it was accepted that a single dose of streptomycin at a dose of 25 mg/kg could cure these cows and was likely to eliminate the renal carrier status [9]. However, some infections can resist this protocol [11,15]. In countries where the use of streptomycin is banned, oxytetracycline, tulathromycin, and ceftiofur have shown some degree of efficacy in treating leptospirosis [11,12].

Fluoroquinolones were not recommended to treat bovine leptospirosis despite their relatively long half-life, good bioavailability, large volume of distribution, and relatively few adverse effects [16]. Ciprofloxacin, norfloxacin, levofloxacin, and gatifloxacin have significantly reduced hamster mortality under laboratory conditions but only administered at high doses and during the leptospiremia phase, thus indicating poor clinical efficacy if the treatment is attempted in cattle [16,17]. In contrast, the recrystallized enrofloxacin derivative: enrofloxacin HClz2H_2_O (enro-C) (Patent MX/a/2013/014605; Mexican Institute of Industrial Protection, Mexico City) was shown to have excellent pharmacokinetic parameters and clinical efficacy in the hamster model of leptospirosis [18,19]. As compared to standard enrofloxacin, enro-C was shown to exhibit superior key pharmacokinetics parameters. Hence, favorable Monte Carlo simulations and clinical outcomes were obtained in dogs [20,21]. In cows, the maximum serum concentration/minimum inhibitory concentrations (C_MAX_/MIC) and area under the curve in 24 h/MIC (AUC_0-24_/MIC) ratios, as well as the Monte Carlo simulations, strongly suggest that enro-C may be useful in the treatment of bovine leptospirosis [22]. Hence, this trial aimed to evaluate the clinical efficacy of enro-C in treating acute and chronic leptospirosis cases in Holstein-Friesian cows in two endemic regions in Mexico, the highlands and a tropical area.

## 2. Material and Methods

### 2.1. Study Design and Animals

All study procedures and animal care activities were carried out following the Institutional Committee for Research, Care, and Use of Experimental Animals of the National Autonomous University of Mexico (UNAM), as per the Official Mexican Regulation (NOM-062-ZOO-1999 (www.fmvz.unam.mx/fmvz/principal/archivos/062ZOO.PDF)). The present experiment was carried out on two groups of animals and in two geographic areas: (1) the Mexican central highlands, characterized by a temperate subhumid climate with summer rains, a range of temperatures from 12 to 16 °C, and annual rainfall of 500–700 mm (HL group); and (2) the tropics in southeastern Mexico, characterized by a hot humid climate with abundant summer rains (1200–3000 mm) and a mean annual temperature of 25 °C (TL group). In the HL group, a total of 111 Holstein-Friesian cows with an average weight of 543 ± 32 kg, aged 2–5 years in intensive farming were included. Of these animals, 13 cows were not vaccinated against any *Leptospira* serovar; 98 cows had immunization records against *Leptospira* spp. with commercial bacterins, Spirovac^®^ (Zoetis, Mexico City, Mexico), Leptosferm-5^®^ (Zoetis, Mexico City, Mexico), and Leptos 10^®^ (Tornel, Mexico City, Mexico). Additionally, all cows were vaccinated against brucellosis, bovine viral diarrhea, and infectious bovine rhinotracheitis. Reproductive management was carried out by artificial insemination. The cows in the TL group were located in an extensive dual-purpose production farm. A total of 38 F1 Zebu x Holstein/Friesians were included with an average weight of 462 ± 47 kg and ages 2–6 years. None of the cows in this latter group were vaccinated against Leptospira spp. but were against brucellosis. The reproductive management in these cows included natural mounting and, occasionally, artificial insemination.

### 2.2. Risk Assessment

In both scenarios, complete individual anamnesis was performed aided by the local veterinarian. Then, individual clinical examinations were conducted, and the risk of having leptospirosis was graded as high or medium, based on the clinical signs commonly associated with leptospirosis, aided by an algorithm that included the following: recent abortion, fetal mummification, hemoglobinuria, flaccid udder, the birth of weak calves, stillbirths, and reduced reproductive efficiency. The scores were: high risk of suffering leptospirosis (score of 10–15); medium risk (score of 6–9); and low risk (<5 points). These latter cows were excluded from this study (Table 1). Then, polymerase chain reaction (PCR) and MAT tests were carried out. In the end, only the cows with a positive PCR were accounted for in the statistical analysis. In addition, the cows with MAT <200 without prior vaccination and <400 with prior immunization were regarded as negative for leptospirosis, and they were left out of the statistical analysis for clinical efficacy. The cows with aggregated pathologies were excluded from this trial.

### 2.3. MAT and PCR Tests

Before treatment, two 7–10 mL blood samples were obtained by direct jugular vein puncture in all of the cows, the serum was separated by centrifugation, and processed within 24 h at the Department of Microbiology (National Autonomous University of Mexico) for MAT [23]. The MAT was repeated 28 days after ending treatment. The agglutinating serovars of the anti-Leptospira antibodies to identify were: Autumnalis, Bataviae, Bratislava, Canicola, Celledoni, Grippotyphosa, Hardjo, Icterohaemorrhagiae, Pomona, Pyrogenes, Tarassovi, and Wolffi. The urine samples were obtained during spontaneous micturition, preferably at mid-void. Approximately 30 mL of urine was collected in 50 mL containers, which were labeled and shipped for PCR tests. Urine samples were analyzed: when initiating the treatment, 5, and 28 days later. DNA extraction was performed using QIAamp DNA Mini kits following the manufacturer’s instructions (Qiagen México S. de R.L. de C.V., Mexico City) with a final volume of 200 μL. Primers were designed as reported by Stoddard [24]: LipL32-45F (5′-AAG CAT TAC CGC TTG TGG TG-3′) and LipL32-286R (5′-GAA CTC CCA TTT CAG CGA tt-3′). The real-time PCR was conducted using a TaqMan PCR kit in a volume of 20 μL containing 400 nM of forwarding primer, 400 nM of reverse primer, 12.5 μL of the kit mix, and 5 μL of DNA clinical extract. The amplification protocol consisted of 5 min at 94 °C followed by 40 cycles (30 s at 94 °C, 30 s at 68 °C, 30 s at 72 °C). After the reaction, the samples were cooled at 40 °C for 120 s.

Because the MAT and PCR results took approximately 7–21 days to be completed, all of the cows initially diagnosed as suffering from leptospirosis were treated. However, only the PCR-positive animals were considered for the statistical analysis in this trial. Additionally, during the treatment and up to 90 days after it, each patient’s clinical and reproductive follow-up was carried out. It was considered that a bacteriological cure was achieved if the result from the PCR test carried out 28 days after treatment became negative.

### 2.4. Group Assignment of Cows

Using a random number generator, a total of 111 cows were assigned to one of two treatments in the HL, i.e., enro-C_1_ with 56 cows and streptomycin/penicillin-G_1_ with 55 cows. Similarly, 38 cows were randomly assigned to two groups in the TL: enro-C_2_ with 19 cows and streptomycin/penicillin-G_2_, also with 19 cows. Enro-C treatment consisted of a daily IM injection for 5 days of a 10% suspension of this antibiotic (dose: 15 mg/kg IM), injecting 10 mL per site in 5–6 injection sites. Treatment with streptomycin/penicillin-G was based on a suspension of 25 mg/kg of streptomycin, plus 15,000 IU/kg of procaine penicillin-G, once a day for 5 days IM, injecting the dose in 4–5 sites not exceeding 10 mL per injection site in the neck and the gluteal muscles.

### 2.5. Enro-C Production

Batches of recrystallized enrofloxacin were prepared as indicated in Patent 472715 (Mexico/Instituto Mexicano de Protección Industrial: IMPI MX/a/2013/014605 and PCT/Mx/ 2014/00192, Mexico City). This process produces enrofloxacin hydrochloride-dihydrate, identified as enro-C. Enrofloxacin with a purity of 99.97% was purchased from Globe Chemicals (Mexico). For the IM injection, a 10% suspension was freshly prepared with sterile water with a pH of 6.5.

### 2.6. Clinical and Bacteriological Cure

The resolution of all clinical signs after treatment with enro-C or streptomycin plus penicillin-G was taken as the endpoint, and it was analyzed using descriptive statistics (mean and standard deviation). Averages of anti-leptospira antibody titers were obtained for each of the serovars (Autumnalis, Bataviae, Bratislava, Canicola, Celledoni, Grippotyphosa, Hardjo, Icterohaemorrhagiae, Pomona, Pyrogenes, Tarassovi, and Wolffi) before and 28 days after treatment. These data were analyzed using a Student’s *t*-test for paired samples [25]. For the independent analyses between treatments in each scenario, analysis variance was performed for the averages of antibodies, and a non-parametric Kruskal–Wallis analysis was performed to determine the bacteriological cure, and *p* < 0.05 was established as the significance level. The recorded resolutions of reproductive parameters were: days to the resolution of retained placenta and metritis; days to the first estrus, the first artificial insemination, the number of inseminations to pregnancy, the interval from insemination to pregnancy, and the gestation percentage on day 90. The pregnancy diagnosis was performed by ultrasonography on day 28 after the last artificial insemination.

## 3. Results

In all, 162 cases of cows graded either as medium or high risk that could have been suffering from leptospirosis were initially considered to enter this trial. However, after obtaining the MAT results, and based on the exclusion criteria already detailed, only 149 cows were finally included. Hence, an initial diagnostic error, based only on clinical signs, of 8.02% was calculated, and this only occurred in the animals graded as medium risk. It is essential to highlight that the cure rates and MAT and PCR data are presented as the sum of medium- and high-risk cows, considering only whether they were enro-C or streptomycin/penicillin-G treated cows.

Table 2 summarizes the total positive cows for Leptospira serovars used in the initial MAT tests and those obtained 28 days after enro-C or streptomycin/penicillin-G administration in the two settings studied. The mean ± SE of the antibody titers against these serovars is presented in Table 3. A significant reduction in anti-leptospira antibody titers was observed in all cases, in both groups, and in both geographical settings (*p* < 0.05 in all cases). Five days after treatment, 49 of the 56 urine samples from the cows in the enro-C_1_ group were PCR negative (87.5%), and 28 days later, this number increased to 52 of 56 (92.85%). The values for the group treated with streptomycin/penicillin-G_1_ were 65.45 and 70.90%, respectively, for days 5 and 28 post-treatment. Similarly, for the tropical scenario in Mexico, the values were 78.94% and 94.73% for enro-C_2_ and 73.68% for streptomycin/penicillin-G_2_ at both sampling times (Table 3). After treatment, the mean antibody titers of serum samples from the enro–C group were significantly lower than those treated with streptomycin plus penicillin-G in the highlands and the tropics. In the highlands, the total of the negative PCR urine samples in the enro-C group was statistically higher than the corresponding values in the group treated with streptomycin/penicillin-G, both on days 5 and 28 (*p* < 0.05 in both cases). However, in the tropics, no significant difference between treatments was observed (Table 4).

Additionally, the follow-up of the reproductive parameters in the treated cases for 90 days allowed us to register a satisfactory reproductive performance (Table 5). The treated cows in the Mexican highlands had gestation rates of 86.53% and 79.06% for enro-C_1_ and streptomycin/penicillin-G_1_, respectively. In the Mexican tropical area pregnancies, the percentages were 88.88% and 87.5% for enro-C_2_ and streptomycin/penicillin-G_2_, respectively. Two cows from the enro-C_1_ group and five from the streptomycin/penicillin-G_1_ group were culled due to the owners’ decision, and reproductive follow-up was impossible.

## 4. Discussion

This trial selected clinical cases based on the combination of reproductive clinical signs and MAT titers. Eventually, confirmation was based on PCR results. To minimize the interference of other factors, some cases were excluded from this study, such as the few cows that required fluids and electrolytes, anti-inflammatory agents, or hormonal treatment, or when other local treatments of the reproductive system were deemed necessary. Based on vaccination history and selected clinical features, other reproductive diseases with similar clinical signs to leptospirosis were excluded, such as brucellosis, neosporosis, bovine viral diarrhea, infectious bovine rhinotracheitis, and campylobacteriosis. Only 11.71% of all treated cows in the highlands and 100% in the tropics (scenario 2) were previously immunized against leptospirosis. Hence, based on OIE recommendations [7], the values ≥200 were taken as an indication of cows without previous vaccination, and the values ≥400 more than 90 days after vaccination were considered seropositive. According to Martins et al. [26], the humoral response occurs rapidly after vaccination, reaching maximum titers on day 60, and from this point onwards, the antibody titers decrease. However, the antibody titration level can vary, depending on the number of serovars included in the bacterin used [27]. In some cases, it may not even induce a humoral response after vaccination [28]. This is particularly true when international reference strains of *Leptospira* are used to manufacture a given bacterin. Such vaccines fail to generate adequate protection against infection caused by native strains of *Leptospira* [6]. Additionally, it is essential to note that the response to vaccination is specific for the serovars included in the utilized bacterin [29]. Hence, the antibody titers >200 of serovars not contained in the bacterin were considered seropositive. MAT and PCR tests were available days and weeks after the initial clinical examination. For this reason, all cows initially diagnosed as suffering from leptospirosis received either of the two treatments. However, only those cows eventually confirmed positive by PCR were quantified for statistical analysis in this trial. It is possible that overcrowding of cows, sanitary management, and control of harmful fauna explain the variations in the antibody titers between the two locations. Extensive management of cows in the tropics reduces contact with contaminated urine from other animals in the herd, and contact with rats, one of the most important reservoirs of this disease.

The dose of enro-C, set at 15 mg/kg, was established based on the high probability of achieving the target ratio of C_MAX_/MIC > 10, based on the Monte Carlo simulations. This PK/PD ratio is essential to optimize the efficacy of enrofloxacin in treating bovine leptospirosis [22]. The choice of streptomycin plus procaine penicillin-G as a reference treatment for leptospirosis in cows was based on recommendations in the literature [9,11]. The high efficacy of streptomycin was accepted, despite its short-term renal elimination and low apparent volume of distribution (<0.2 L/kg) [30]. It appears that streptomycin may be useful in cases of leptospiremia or when an accumulation of *Leptospira* spp. microorganisms occur in the proximal renal tubule, but streptomycin is unlikely to reach therapeutic concentrations in the reproductive tract [31,32], a widespread chronic presentation of this disease in cattle [33]. Other antibacterial drugs were shown to possess some degree of efficacy, for example: amoxicillin [34], oxytetracycline, tilmicosin, and ceftiofur [11]. However, they are still not as effective as streptomycin plus penicillin-G. In contrast, fluoroquinolones are not particularly effective in treating leptospirosis in veterinary medicine, perhaps due to the gap that exists between the drug concentrations achieved at tissue level and the necessary MIC required. For example, it was stated that enrofloxacin requires from 1 to 4 µg/mL as the MIC for Leptospira spp., and a much higher dose should be administered if optimal efficacy is sought [16,17,35]. In comparison, enro-C is highly effective against *Leptospira* spp. in the hamster model [36] and clinical cases in dogs [21]. However, neither enro-C nor other fluoroquinolones were tested in cattle. In this trial, the motivation to test the use of enro-C to treat cases of leptospirosis in cows was based on its already mentioned improved PK/PD ratio, given the volume of distribution of enrofloxacin in cows (2.5–4 L/kg) [30,37]. Indeed, achieving therapeutic concentrations of enrofloxacin in the renal and reproductive tract when using enro-C requires confirmation, but the cure rates observed in this trial suggest it is possible. The decrease in the antibody titers observed after treatment with 15 mg/kg of enro-C follows the patterns reported by Gerritsen and colleagues [10] after administering a dose of 25 mg/kg/day of streptomycin and complies well with the previously reported efficacy of enro-C to treat dogs [21]. Hence, it is safe to assume that this trend toward reduction in antibody titers is associated with a decrease in or absence of bacteria, as Carrascosa et al. [36] demonstrated in the hamster leptospirosis model. Additionally, the 92.85 and 90.47% PCR-negativity in urine samples 28 days after the end of the treatment with enro-C, as well as the percentage of pregnancies reached in this trial (86.53% and 88.88% for the HL and TL, respectively), suggest that a bacteriological cure with enro-C is possible. Further tests on cattle should be carried out to support this claim.

From an ethical point of view, it is questionable and very costly to induce leptospirosis in healthy cows [13]. Therefore, clinical trials during field outbreaks become essential. Particularly, multicenter studies under different environments and management conditions may strengthen or weaken the evidence of the efficacy of an antibacterial drug in a test. This premise was attempted for enro-C in this trial, and the results concede a high efficacy for enro-C in treating leptospirosis in cows. Additionally, no adverse effects were observed, even at the injection sites, as the 10% suspension of enro-C has a pH of 6.5 and was neither an irritant nor painful when injected. Additionally, it is essential to note that, although antibiotic therapy is a significant part of the strategy to control outbreaks of bovine leptospirosis, vaccination and the implementation of sanitary measures must be carefully assessed to control this disease. Finally, it is worth pointing out that no streptomycin-based product in Mexico is designed to treat bovine leptospirosis. Despite the pharmacological evidence available worldwide against procaine penicillin-G plus streptomycin combinations [30,38], all the pharmaceutical preparations of streptomycin found in Mexico contain penicillin-G. Hence, the reference treatment set in this trial was necessarily based on this combination.

## 5. Conclusions

A daily IM injection of 5 mg/kg enro-C for 5 days in cows with leptospirosis is highly effective in the treatment of bovine leptospirosis, i.e., 92.85% and 94.73% of clinical cures were demonstrated by PCR of urine samples 28 days after treatment in either the highlands or the tropics, respectively. A reproductive clinical follow-up guarantees the proposal of a bacteriological cure. This is the first report of a fluoroquinolone derivative being highly effective in treating leptospirosis in cattle. Consequently, the use of enro-C as an option to treat clinical leptospirosis is proposed, mainly when the use of streptomycin–penicillin-G is not an option.

## Figures and Tables

**Table 1 animals-12-02358-t001:** Scoring system to determine the final risk level of a cow suffering from leptospirosis according to the clinical signs and the antibody titers against *Leptospira* spp. by the microscopic agglutination test (MAT). High risk of suffering from leptospirosis (score of 10–15), medium risk (score of 6–9), and low risk (<5 points).

Clinical Signs	Score
Abortion	3
Fetal mummification	3
Hemoglobinuria	3
Flaccid udder	2
Weak-born calf or Stillbirth	2
Previous abortions	1
Infertility	1
Increased calving interval	1
**MAT**	
Vaccinated	
>1:800	4
1:400	2
1:200	0
1:100	0
Unvaccinated	
>1:400	4
1:200	2
1:100	1
Not reactive	0

**Table 2 animals-12-02358-t002:** Serodiagnosis of leptospirosis by microscopic agglutination test (MAT) before and after treatment with either enro-C or streptomycin/penicillin-G.

	Enro-C_1_		Streptomycin/Penicillin-G_1_		Enro-C_2_		Streptomycin/Penicillin-G_2_	
Serovar	before Tx	28 d after Tx	before Tx	28 d after Tx	before Tx	28 d after Tx	before Tx	28 d after Tx
Autumnalis	1	1	4	3	8	2	5	2
Bataviae	3	1	8	6	2	0	0	0
Bratislava	43	33	36	32	14	6	9	7
Canicola	37	26	41	30	10	4	11	7
Celledoni	2	1	1	1	0	0	2	1
Grippotyphosa	32	14	13	10	7	4	4	3
Hardjo	36	18	37	24	11	8	13	9
Icterohaemorrhagiae	8	1	15	8	4	2	7	4
Pomona	32	20	24	19	4	1	6	3
Pyrogenes	12	3	18	12	2	1	1	1
Tarassovi	1	0	0	0	1	0	0	0
Wolffi	38	26	31	24	9	7	12	9

**Table 3 animals-12-02358-t003:** Antibody titers from the microscopic agglutination test (MAT) and polymerase chain reaction (PCR) from urine samples in cows affected by *Leptospira* spp. and treated either with enrofloxacin hydrochloride-dihydrate (enro-C) or streptomycin plus procaine penicillin-G, in two geographical settings: the highlands (1) and the tropics in Mexico (2).

Treatment	Mean ± EE Antibody Titers		Real-Time PCR Negative/Positive (%)		
	before Tx	28 d after Tx	before Tx	5 d after Tx	28 d after Tx
enro-C_1_	704 ± 71.94 ^a^	329.79 ± 29.27 ^b^	0/56 (100) ^a^	49/7 (87.5) ^b^	52/4 (92.85) ^b^
Streptomycin/Penicillin-G_1_	615 ± 39.76 ^a^	430.16 ± 33.15 ^b^	0/55 (100) ^a^	36/19 (65.45) ^b^	39/16 (70.90) ^b^
enro-C_2_	271.05 ± 36.93 ^a^	88.15 ± 10.50 ^b^	0/19 (100) ^a^	15/4 (78.94) ^b^	18/1 (94.73) ^b^
Streptomycin/Penicillin-G_2_	288.57 ± 32.20 ^a^	158.57 ± 19.70 ^b^	0/19 (100) ^a^	14/5 (73.68) ^b^	14/5 (73.68) ^b^

^a,b^ Different letters indicate statistically significant differences between values in a row (*p* < 0.05).

**Table 4 animals-12-02358-t004:** Antibody titers by the microscopic agglutination test (MAT) and polymerase chain reaction (PCR) of urine samples in cows affected by *Leptospira* spp. after treatment with enrofloxacin hydrochloride-dihydrate (enro-C) or streptomycin plus procaine penicillin-G in the highlands (1) and a tropical area in Mexico (2).

Treatment	Mean ± EE Antibody Titers	Real-Time PCR Negative/Positive (%)	
		5 d	28 d
enro-C_1_	329.79 ± 29.27 ^a^	49/7 (87.5) ^a^	52/4 (92.85) ^a^
Streptomycin/penicillin-G _1_	430.16 ± 33.15 ^b^	36/19 (65.45) ^b^	39/16 (70.90) ^b^
enro-C_2_	88.15 ± 10.50 ^a^	15/4 (78.94) ^a^	18/1 (94.73) ^a^
Streptomycin/penicillin-G _2_	158.57 ± 19.70 ^b^	14/5 (73.68) ^a^	14/5 (73.68) ^a^

^a,b^ Different letters in the same column show a statistically significant difference (*p* < 0.05).

**Table 5 animals-12-02358-t005:** Mean ± 1SD values of reproductive parameters observed until day 90 after ending the treatment of cows with leptospirosis with either the IM administration of 15 mg/kg of enrofloxacin hydrochloride-dihydrate (enro-C) for 5 d or 25 mg/kg of streptomycin/penicillin-G procaine (15,000 IU/kg) for 5 d. Two geographic scenarios in Mexico are presented: the highlands with Holstein/Friesian cows and a tropical area in Mexico with F1 Zebu x Holstein/Friesian cows.

Reproductive Parameters	Enro-C_1_	Streptomycin/Penicillin-G_1_	Enro-C_2_	Streptomycin/Penicillin-G_2_
Resolution of metritis (days)	44.29 ± 21.47	58.63 ± 16.78	44 ± 11.31	---
Retained placenta (days) *	42.16 ± 14.41	45.5 ± 15.94	30.66 ± 9.71	33.25 ± 10.65
First estrus (days) ^†^	39 ± 15.42	46.62 ± 18.31	34.77 ± 11.23	39.62 ± 12.10
First artificial insemination (days) ^‡^	47 ± 20.76	53.53 ± 16.97	---	---
First artificial insemination to pregnancy (days)	22.5 ± 34.48	26.33 ± 28.32	---	---
Services per conception	1.88 ± 1.03	2.04 ± 1.13	---	---
Pregnancy (%) ^§^	86.53	79.06	88.88	87.5

* Time from placental retention to first estrus, ^†^ Interval from calving, abortion, or treatment of mummification to the first detected estrus, ^‡^ Interval from birth, abortion, or treatment of mummification to the first artificial insemination. ^§^ Total pregnant cows after abortion, fetal mummification, placental retention, or metritis. It was not feasible to compare the cows in this study with general herd data or to establish a control group affected with leptospirosis without receiving treatment.

## Data Availability

All datasets generated for this study are included in the manuscript. Supplementary files are available upon reasonable request from the corresponding author.

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
