# Peer review of "Treatment of Bovine Leptospirosis with Enrofloxacin HCl 2H2O (Enro-C): A Clinical Trial"

_animals, 2022, doi:10.3390/ani12182358_

Round 1

Reviewer 1 Report

Dear author

Your manuscript ID 1860291, entitled TREATMENT OF BOVINE LEPTOSPIROSIS WITH ENROFLOXACIN HCl • 2H2O (enro-C). A CLINICAL TRIAL provides a new tool for the treatment of bovine leptospirosis. However, I would like you to consider the following comments to clarify some concepts:

2.3. MAT and PCR tests,

Line 141: How were urine samples collected? please describe

Line 166: ......plus 15,000 IU/kg of procaine penicillin G,g once a day for 5 days IM. 

3. Results

Line 215: Change 8 for 28

Tables 2 and 3: Arrange all column headers please and arrange the values to the corresponding rows to make it clearer.

4. Discussion

If FQs are not effective in treating leptospirosis, to what do you attribute their results?

Your study would be much more robust if you accompanied it with quantification of enro-C in the reproductive tract and urine.

Author Response

2.3. MAT and PCR tests,

QUERY: Line 141: How were urine samples collected? please describe

ANSWER: Done.

QUERY: Line 166: ......plus 15,000 IU/kg of procaine penicillin G,g once a day for 5 days IM. 

ANSWER: Done, thank you!

  1. Results

QUERY: Line 215: Change 8 for 28

ANSWER: Done

QUERY: Tables 2 and 3: Arrange all column headers please and arrange the values to the corresponding rows to make it clearer.

ANSWER: Done, thanks for your suggestion.

  1. Discussion

QUERY: If FQs are not effective in treating leptospirosis, to what do you attribute their results?

ANSWER: line 284-289. The following argument has been added. Thank you for the suggestion.

In contrast, fluoroquinolones are not particularly effective in treating leptospirosis in veterinary medicine, perhaps due to the gap that exist between drug concentrations achieved at tissue level, and the necessary MIC required. For example, it has been stated that enrofloxacin requires from 1 to 4 µg/mL as MIC for Leptospira spp. and 10-12 times higher if optimal efficacy is sough for [16,17,35].

QUERY: Your study would be much more robust if you accompanied it with quantification of enro-C in the reproductive tract and urine.

ANSWER: Thank you for your comment. Although costly, we consider your view as a brilliant suggestion to demonstrate whetehr or not high concentrations of enrofloxacin from enro-C are achieved in the reproductive tract. We will pursue this idea in future studies, and reinforce this way the efficacy of enro-C for the treatment of Leptospirosis in cattle.

Reviewer 2 Report

A current study was carried out to determine the effect of recrystallized enrofloxacin HCl-2H2O (author proprietary drug which they  named it as enro-C) on the treatment of bovine leptospirosis. Authors compared the effect of enrofloxacin HCl-2H2O treatment with streptomycin/penicillin-G in treating bovine leptospirosis. Authors concluded that enrofloxacin HCl-2H2O treats this disease (e.g. bovine leptospirosis) better as compared to streptomycin/penicillin-G.

This study is a repetition of the author’s other study where they investigated the efficacy of this drug (enrofloxacin HCl-2H2O) in treating leptospirosis in sheep and dogs (but unique for cattle)

Overall, the study is well designed and things need to be addressed.

1.      C1 and C2 treatment; G1 and G2 treatment are not clear in material and methods while it is found during reading the results (C1/G1 treatment for highlands and C2/G2 treatment for tropic in Mexico. It will be better if the author make clear at material and method

2.      Why the values of titer are different in Tabl3 3 vs Table 4 (at the end of treatment) for example 328±29.32 vs 329.79±29.27 and 441±34.23 vs 430.16±33.15.

3.      Author concluded that the gestation situation was improved with enro-C following treatment however there is no data before the treatment to compare that effect.

While looking at the leptospira titer:  C2/G2 treatment (tropic in Mexico) seems to have less titer: what would be a possible cause for that? As this information will be useful for leptospira management across different geographical locations

Author Response

QUERY: C1 and C2 treatment; G1 and G2 treatment are not clear in material and methods while it is found during reading the results (C1/G1 treatment for highlands and C2/G2 treatment for tropic in Mexico. It will be better if the author make clear at material and method

ANSWER: Thank you for your comment, the information was reviewed and corrected.

QUERY: Why the values of titer are different in Tabl3 3 vs Table 4 (at the end of treatment) for example 328±29.32 vs 329.79±29.27 and 441±34.23 vs 430.16±33.15.

ANSWER: First of all, we thank you for your observation. The data was reviewed and corrected and the statistical analysis was redone. We may add that the updated data did not modify the response in a statistically significant manner. Thanks again!

QUERY: Author concluded that the gestation situation was improved with enro-C following treatment however there is no data before the treatment to compare that effect.

ANSWER: Thank you for your insightful observation.

Table 5 shows the clinical signs of the cows affected with leptospirosis at the beginning of the study and their evolution after receiving one of the two treatments. It was not feasible to compare the cows in this study with general herd data or to establish a control group affected with leptospirosis without receiving treatment. So we have clarified this in table 5.

QUERY: While looking at the leptospira titer:  C2/G2 treatment (tropic in Mexico) seems to have less titer: what would be a possible cause for that? As this information will be useful for leptospira management across different geographical locations

ANSWER: After an exhaustive review, the authors suggest that overcrowding, sanitary management and control of harmful fauna is a cause of the variations in antibody titers between the two locations. Extensive management reduces contact with contaminated urine from other animals in the herd, as well as contact with rats as the main reservoir of the disease. This has been added to the discussion and we greatly appreciate your observation. Please refer to lines 273-277